# PRIOR DISTRIBUTION AND MODEL CONFIDENCE

## ABSTRACT

This paper investigates the impact of training data distribution on the performance of image classification models. By analyzing the embeddings of the training set, we propose a framework to understand the confidence of model predictions on unseen data without the need for retraining. Our approach filters out low-confidence predictions based on their distance from the training distribution in the embedding space, significantly improving classification accuracy. We demonstrate this on the example of several classification models, showing consistent performance gains across architectures. Furthermore, we show that using multiple embedding models to represent the training data enables a more robust estimation of confidence, as different embeddings capture complementary aspects of the data. Combining these embeddings allows for better detection and exclusion of out-of-distribution samples, resulting in further accuracy improvements. The proposed method is model-agnostic and generalizable, with potential applications beyond computer vision, including domains such as Natural Language Processing where prediction reliability is critical.

## 1 INTRODUCTION

In recent years, we have seen unprecedented development of deep learning models and an extension of the problems that can be effectively solved by them. They have dramatically advanced AI capabilities, but their training paradigm imposes fundamental constraints. Leading companies, such as OpenAI, have utilized nearly all accessible data to build these models, which require immense computational resources and time (Brown et al., 2020). However, this approach presents major limitations, such as the difficulty in optimizing model architectures and the exclusivity of achieving such results to a handful of research institutions and corporations. Consequently, most scientists and engineers are limited to using pre-trained models without the ability to modify or improve them (Bommasani et al., 2021). Shortly after deployment, numerous instances of erroneous outputs, termed 'hallucinations,' were identified in these models. These are cases where the response sounds convincing and well-structured but conveys false or unverified information (Huang et al., 2023). A significant challenge with these errors is the difficulty in detecting their root causes through automated means, even though users can often recognize them with ease. This poses a serious risk in industrial environments, where such inaccuracies can trigger unforeseen system behaviors or failures in AI-driven technologies.

Although understanding the factors behind such behavior in AI systems is challenging due to their inherent complexity, models of that type (such as Huawei's PanGu-$\Sigma$) often contain over 1 trillion parameters (Ren et al., 2023). Mathematical structures of this magnitude are extremely difficult to analyze from both theoretical and technical perspectives (Hudson et al., 2024).

The field of computer vision continues to be highly suitable for theoretical exploration. Unlike NLP, where constructing representative datasets for industrial-level models is difficult due to the enormous data requirements, computer vision faces fewer such challenges. As one of the earliest domains enhanced by deep learning, it now benefits from a rich collection of well-curated datasets ideal for research purposes (Wan, 2024).

Consequently, using computer vision models to investigate the nature of "hallucinations" appears more reasonable. As is well-known, training data is a primary factor influencing model performance, as it fundamentally determines the model's scope and applicability (Chen et al., 2024). Therefore,

understanding the distribution of training data is crucial for assessing a model's predictability (Győri et al., 2022; Dabiri et al., 2023).

This paper presents a foundational approach for designing algorithms to detect hallucinations. The core idea is to leverage the latent space representation of training data to assess the model's predictive capabilities on specific instances. While the current study focuses on deep learning models in computer vision, the proposed methodology is general enough to be extended to other machine learning and deep learning domains, including natural language processing.

## 2 RELATED WORK

There are several approaches developed to mitigate "hallucinations" without recreating the model.

A practical first step is to adapt the model's parameters for a given task using tuning strategies like fine-tuning or alignment (Zhang et al., 2023; Wu et al., 2025; Lialin et al., 2023; Prottasha et al., 2024) . However, these techniques often require significant amounts of extra training data and may still lack dependable reliability.

Another solution is to mitigate hallucinations by using RAG (Retrieval-Augmented Generation) techniques, which integrate information retrieval with generation. This allows the model to reference real-world information, making the output more accurate and reliable (Béchard & Ayala, 2024). Another option is to apply statistical and information-theoretic methods to detect hallucinations, aiming to flag outputs that deviate from expected or reliable patterns (Farquhar et al., 2024; Li, 2025). While these methods focus on analyzing the output of deep learning networks, they often overlook the distribution of the training data. Yet, the structure and spread of training data across complex feature spaces can critically impact a deep learning model's performance (Hammoudeh & Lowd, 2024; Lin et al., 2022). Accurately measuring and modeling input data distribution is a crucial strategy for enhancing model performance in machine learning. Numerous methods have been developed to address this challenge effectively (Hammoudeh & Lowd, 2024; Tamang et al., 2025).

First, training data is formally described and cataloged using tools like Data Cards, and visualized through techniques such as t-SNE or UMAP applied to CLIP or DINO embeddings. Mislabeled instances can be identified and corrected with Confident Learning (Northcutt et al., 2021), while Dataset Cartography (Swayamdipta et al., 2020) enables categorization of data points into "easy," "hard," or "ambiguous" classes. Second, distribution shift and coverage are measured. To detect samples outside the training distribution, OOD (out-of-distribution) methods like ODIN (Liang et al., 2018), Mahalanobis Distance (Lee et al., 2018), Energy-based OOD (Liu et al., 2020), and Open OOD (Yang et al., 2022) are used. As an opposite approach, we can start from the defined distribution of the training data and try to fit the data to it. The Core Set method (Sener & Savarese, 2018) selects diverse representatives to cover the input space. To select uncertain and diverse examples, the BADGE method (Ash et al., 2020), based on gradient embeddings, can be used. If you have a set of specific features, the Submodular Selection method (Wei et al., 2015) ensures their coverage. In the case of class imbalance, the Focal Loss method (Lin et al., 2017) reweights samples from imbalanced classes. Methods like LAION (Schuhmann et al., 2021) and DataComp (Gadre et al., 2023) demonstrate that targeted filtering toward the desired distribution can significantly outperform simply increasing dataset size in terms of accuracy.

In addition, several methods focus on evaluating training data by analyzing the contribution of its individual components. Influence Functions (Koh & Liang, 2017) and TracIn (Pruthi et al., 2020) help quantify the impact of specific training samples on model predictions, making it possible to identify non-representative data points that may degrade performance. Other evaluation approaches, such as VQA v2 (Goyal et al., 2017), aim to reduce shortcut learning by rebalancing answer priors, thereby encouraging models to rely on visual information. Similarly, ObjectNet is designed to assess generalization capabilities under varying viewpoints and backgrounds.

## 3 METHODS

This manuscript presents a method for understanding the influence of the prior data distribution on the confidence of prediction, independent of the predictive signal of the model.

To investigate the effect of data distribution on model confidence, we selected several models trained on the same dataset. Specifically, all models were trained using the ImageNet-1K dataset (Deng et al., 2009), ensuring a consistent distribution of training data between experiments. In order to understand the difference between the performance of our algorithm, we intentionally took several models from different classes with different numbers of parameters. The chosen models are: ResNet type models(Resnet-101 and Resnet-50 (He et al., 2016)), Visual Transformers (Touvron et al., 2021) (Deit-T, Deit-S and Deit-B) and a relatively small model ShuffleNet-V2 (Ma et al., 2018). The brief description of the models is shown in Table 1.

Table 1: Overview of the models used in our experiments. All models were trained on the ImageNet-1K dataset to ensure consistent data distribution.

| Model | Training Dataset | Architecture Type | Reference | # Parameters |
|---|---|---|---|---|
| ResNet-101 | ImageNet-1K | Convolutional (CNN) | (He et al., 2016) | 44.5M |
| ResNet-50 | ImageNet-1K | Convolutional (CNN) | (He et al., 2016) | 25.6M |
| ShuffleNet-V2 | ImageNet-1K | Lightweight CNN | (Ma et al., 2018) | 2.3M |
| DeiT-Tiny | ImageNet-1K | Vision Transformer (ViT) | (Touvron et al., 2021) | 5.7M |
| DeiT-Small | ImageNet-1K | Vision Transformer (ViT) | (Touvron et al., 2021) | 22.1M |
| DeiT-Base | ImageNet-1K | Vision Transformer (ViT) | (Touvron et al., 2021) | 86.4M |

To understand the data distribution on which the models were trained, we analyzed the structure of the ImageNet-1K training set by computing image embeddings using several pretrained models. Specifically, we selected a diverse set of architectures to generate embeddings for each image in the dataset. These embedding-based representations allowed us to examine the statistical properties of the data in different feature spaces. The analysis provides insight into how various models perceive and encode the visual information present in the training distribution. It is worth mentioning that we took only the "train" part of the set since exactly on this set the models in Table 1 were trained. For the embedding analysis, we chose the following models: The Dino-V1 model (Caron et al., 2021), Dino-V2 models (Oquab et al., 2023) with different embedding sizes (384, 768, 1024) and the MobileNet-V2 model (Sandler et al., 2018). Our choice of embedding models was motivated by two key considerations: the model size and the nature of the data used for pre-training. To assess the sensitivity of our algorithm to embedding dimensionality, we evaluated its performance on closely related models with varying embedding sizes. In addition, we were interested in understanding the effects of the data on which the embedding models were trained. We include a brief description of the embedding models in Table 2.

Table 2: Description of the models and their embedding sizes.

| Model | Training Data | Embedding Size | Reference |
|---|---|---|---|
| DINO-V1 ViT-S/16 | ImageNet-1K (self-supervised) | 384 | (Deng et al., 2009) |
| DINO-V1 ViT-B/16 | ImageNet-1K (self-supervised) | 768 | (Deng et al., 2009) |
| DINO-V1 ViT-B/8 | ImageNet-1K (self-supervised) | 768 | (Deng et al., 2009) |
| DINO-V2 ViT-S/14 | 142M curated dataset (no labels) | 384 | (Oquab et al., 2023) |
| DINO-V2 ViT-B/14 | 142M curated dataset (no labels) | 768 | (Oquab et al., 2023) |
| MobileNet-V2 | ImageNet-1K (supervised) | 1000 | (Sandler et al., 2018) |

In order to investigate the effect of the prior data distribution on the prediction capabilities of the classification models, we have chosen two distinct datasets: ObjectNet(Barbu et al., 2019) and imageNet-V2(Recht et al., 2019). We divided the imageNet-V2 into train and internal test sets with a split (75% / 25%) and left the ObjectNet dataset as the external test set. We used only Object-Net classes that coincide with the classes in the original ImageNet1K dataset (which gave us $10\,121$ images). We used the train dataset to find the optimal parameter: the number of neighbors $N$. We divided the test sets (external and internal) into three subsets grouped by the labels. So, images with the same label can only be present in one subset. The goal of this division is to report the variation of the metrics.

For the calculation of embeddings, we used cosine similarity (Manning et al., 2008), which is widely regarded as an effective metric to identify neighbors in high-dimensional spaces. For storing and querying the embeddings, we used ChromaDB (Team, 2023) with ClickHouse (ClickHouse, 2016)

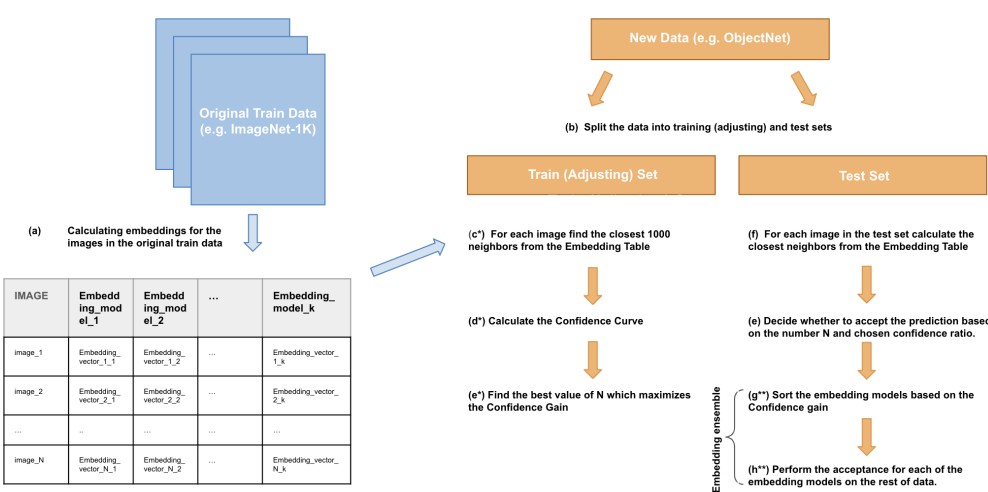

Figure 1: The scheme of the experiment. With (*) the training (adjusting) steps are indicated. With (**) supplementary combination predictions steps are shown

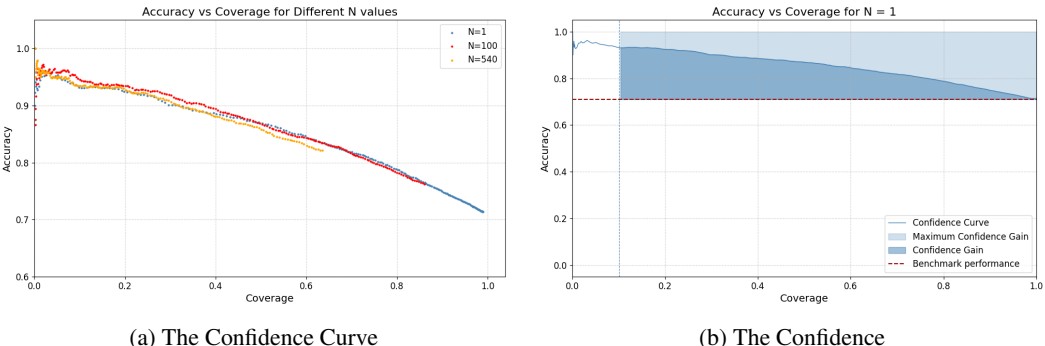

(a) The Confidence Curve          (b) The Confidence

Figure 2: The Confidence Curves and Confidence Gain for the ResNet-101 & DINO-V2 ViT-B/14 .

as the backend server to enable efficient retrieval. All calculations were performed on a MacBook with an Apple M1 chip.

## 4 EXPERIMENTS

Confidence estimation was conducted on both the training and test sets. In the training set, we adjusted the parameter $N$ to optimize performance. The procedure for adjusting confidence in the training set involves the following steps. An illustration of both the training and test pipelines is provided in Figure 1.

1. For each image in the original training dataset ImageNet-1K (let us call it Base Image Set), we calculate the embeddings for embedding models presented in Table 2 using cosine similarity. We store these embeddings in the Base Embedding Database. Although the Base Image Set serves as a training set for the models we investigated, we do not fine-tune its parameters. Therefore, the word "training" could be misleading.

2. For the training dataset (a subset of ImageNet-V2), we compute the embeddings for each image and identify 1000 closest neighbors from a base embedding database. This allows us to incorporate information about the local distribution of embeddings. We choose this

particular number to ensure that we fully describe the neighbors of a particular image. We refer to this collection as the Train Embedding Set. The term "train" emphasizes that this dataset is used to tune the parameter $N$, which is then kept fixed during the evaluation on the test sets.

3. For each image in the train image set, we calculate and store the predictions of the classification model.

4. For each image in the train set, we calculate the following statistics. We use the parameters $L_{threshold}$ and $N$ as the threshold value for the distance of the neighbors and the necessary number of neighbors within the proximity $L_{threshold}$. Should the quantity of neighboring elements for the embedding of the image exceed $N$ within the $L_{threshold}$ vicinity, we accept the model's prediction as valid. Conversely, if this threshold is not met, we conclude that the model lacks sufficient confidence to make a prediction. We calculate the accuracy for all images where the confidence is enough. Let us call the percentage of approved images the *Coverage*. The Confidence Curve would be the dependence of the accuracy upon the coverage. The examples of Confident Curves for ResNet-101 & DINO-V2 ViT-B/14 for three different $N$ are presented in Figure 2(a). In Figure 3(b) *Maximum Confidence Gain* and *Confidence Gain* are shown for the same embedding / model pair, and $N = 1$. For more details on how the values are calculated, please see Appendix 1.

5. We find the values of $N$ that maximize the *Confidence Gain* (the ratio between the *Maximum Confidence Gain* and the *Confidence Gain*). We use the value of neighbors for the test sets.

6. We repeated steps 3,4,5 for each of the classification models.

In Figure 4, the results of the adjustment (or training) are shown. On the x-axis, the number of neighbors $N$ is presented, and on the y-axis, the calculated *Normalized Confidence Gain* is depicted.

Table 3: $N$ and best *Normalized Confidence Gain* for each classification model across embedding models.

| Classification Model | DINO-V1 ViT-B/16 | DINO-V1 ViT-B/8 | DINO-V1 ViT-S/16 | DINO-V2 ViT-B/14 | DINO-V2 ViT-S/14 | MobileNet-V2 |
|---|---|---|---|---|---|---|
| deit_base_patch16_224 | 1 / 0.398 | 1 / 0.441 | 1 / 0.368 | 24 / **0.480** | 4 / 0.419 | 1 / 0.194 |
| deit_small_patch16_224 | 1 / 0.408 | 1 / 0.449 | 1 / 0.379 | 7 / **0.472** | 2 / 0.427 | 1 / 0.199 |
| deit_tiny_patch16_224 | 1 / 0.402 | 1 / 0.433 | 1 / 0.379 | 23 / **0.442** | 4 / 0.406 | 1 / 0.218 |
| ResNet-101 | 1 / 0.394 | 1 / 0.434 | 1 / 0.364 | 60 / **0.477** | 4 / 0.417 | 1 / 0.195 |
| ResNet-50 | 1 / 0.407 | 1 / 0.448 | 1 / 0.380 | 24 / **0.487** | 2 / 0.426 | 1 / 0.209 |
| shufflenet_v2_x1_0 | 1 / 0.418 | 1 / **0.436** | 1 / 0.401 | 7 / 0.417 | 4 / 0.395 | 1 / 0.244 |

Table 4: The Normalized Confidence Gain for different pairs of embedding/classification models and for the combination of embedding models. Benchmark accuracy is shown for reference. All values are mean ± std across three label subsets for internal and external test sets.

| | | | | Internal Test | | | |
|---|---|---|---|---|---|---|---|
| Classification Model | Benchmark Accuracy | DINO-V1 ViT-B/16 | DINO-V1 ViT-B/8 | DINO-V1 ViT-S/16 | DINO-V2 ViT-B/14 | DINO-V2 ViT-S/14 | MobileNet-V2 | Combination |
| deit_base_patch16_224 | 0.696 ± 0.014 | 0.390 ± 0.052 | 0.414 ± 0.048 | 0.357 ± 0.035 | **0.449 ± 0.077** | 0.404 ± 0.046 | 0.194 ± 0.046 | 0.361 ± 0.058 |
| deit_small_patch16_224 | 0.666 ± 0.019 | 0.376 ± 0.068 | 0.403 ± 0.071 | 0.343 ± 0.062 | **0.431 ± 0.076** | 0.376 ± 0.058 | 0.189 ± 0.058 | 0.320 ± 0.060 |
| deit_tiny_patch16_224 | 0.582 ± 0.037 | 0.376 ± 0.076 | 0.388 ± 0.079 | 0.353 ± 0.078 | **0.408 ± 0.070** | 0.380 ± 0.067 | 0.202 ± 0.069 | 0.366 ± 0.067 |
| ResNet-101 | 0.687 ± 0.025 | 0.345 ± 0.074 | 0.369 ± 0.066 | 0.308 ± 0.072 | **0.410 ± 0.106** | 0.366 ± 0.058 | 0.169 ± 0.065 | 0.316 ± 0.071 |
| ResNet-50 | 0.676 ± 0.014 | 0.377 ± 0.050 | 0.408 ± 0.049 | 0.347 ± 0.050 | **0.446 ± 0.075** | 0.398 ± 0.044 | 0.189 ± 0.068 | 0.344 ± 0.061 |
| shufflenet_v2_x1_0 | 0.538 ± 0.022 | 0.363 ± 0.056 | 0.374 ± 0.063 | 0.345 ± 0.056 | **0.379 ± 0.053** | 0.357 ± 0.046 | 0.220 ± 0.050 | 0.351 ± 0.046 |

| | | | | External Test | | | |
|---|---|---|---|---|---|---|---|
| Classification Model | Benchmark Accuracy | DINO-V1 ViT-B/16 | DINO-V1 ViT-B/8 | DINO-V1 ViT-S/16 | DINO-V2 ViT-B/14 | DINO-V2 ViT-S/14 | MobileNet-V2 | Combinations |
| deit_base_patch16_224 | 0.269 ± 0.059 | 0.079 ± 0.109 | 0.132 ± 0.120 | 0.059 ± 0.107 | **0.214 ± 0.153** | 0.189 ± 0.149 | 0.044 ± 0.084 | 0.141 ± 0.130 |
| deit_small_patch16_224 | 0.229 ± 0.045 | 0.072 ± 0.092 | 0.120 ± 0.099 | 0.055 ± 0.082 | **0.188 ± 0.125** | 0.170 ± 0.119 | 0.037 ± 0.063 | 0.086 ± 0.087 |
| deit_tiny_patch16_224 | 0.152 ± 0.041 | 0.073 ± 0.077 | 0.102 ± 0.085 | 0.058 ± 0.072 | 0.119 ± 0.092 | **0.119 ± 0.093** | 0.040 ± 0.057 | 0.106 ± 0.090 |
| ResNet-101 | 0.304 ± 0.057 | 0.069 ± 0.112 | 0.131 ± 0.112 | 0.049 ± 0.101 | **0.238 ± 0.144** | 0.211 ± 0.144 | 0.036 ± 0.080 | 0.147 ± 0.123 |
| ResNet-50 | 0.293 ± 0.056 | 0.069 ± 0.102 | 0.127 ± 0.106 | 0.047 ± 0.101 | **0.243 ± 0.130** | 0.211 ± 0.135 | 0.035 ± 0.082 | 0.136 ± 0.114 |
| shufflenet_v2_x1_0 | 0.117 ± 0.027 | 0.062 ± 0.064 | 0.079 ± 0.065 | 0.047 ± 0.061 | **0.096 ± 0.069** | 0.090 ± 0.068 | 0.038 ± 0.048 | 0.083 ± 0.066 |

We can see that the highest *Normalized Confidence Gain* value is achieved for all models in besides the ShuffleNet-V2 by the DINO-V2 ViT-B / 14 embedding model, which is so far the strongest model among embedding models in terms of the number of parameters. Interestingly, the second version of the DINO models outperforms the first version. In table 3, the best parameter $N$ and the corresponding *Normalized Confidence Gain* are presented.

For the test pipeline, we use the number of parameters $N$, which maximizes the *Normalized Confidence Gain* for the specific pair of the classification model and the embedding model. We use this value to calculate the metric for the Internal Test and External Test sets. The metrics and their standard deviations (for 3 subsets of each of the External and Internal Tests) are reported in Table 4. We can observe that the highest value of *Normalized Confidence Gain* is reached almost for all cases by Dino-v2 ViT-B14. As mentioned earlier, this is quite logical and can be explained by the model's higher capacity and the higher resolution of the input images.

The next logical step would be to combine several embedding models (or use an approach similar to the ensemble of models, as in (Dietterich, 2000)), which has been done using greedy heuristics. We fix the same coverage for each embedding model. We take the model with the highest *Normalized Confidence Gain* and make predictions. We accept a portion of the predictions and leave the rest to other models. It should be clear that the total coverage for all embedding models is slightly higher than the parameter *coverage* for individual models, but is not fixed. The results of the experiments (accuracy vs. coverage) are presented in Figure 5. The *Normalized Confidence Gain* for the combination of embedding models is shown in Table 4 in the column "Combination". The observed *Confidence Gain* for the combination of embedding models is lower than that of the best-performing individual model. In particular, at very low coverage values for the internal dataset, the accuracy of all models converges to similar levels. This suggests that the underlying data distribution plays a more decisive role than the intrinsic performance of the models. A plausible explanation is that, within the test datasets, certain images are located in close proximity (in the embedding space) to images from the training set. Consequently, under such conditions, the specific choice of the classification model becomes less critical. For the external dataset, the same conclusions cannot be drawn. This is probably related to the fact that the images in the external dataset (ObjectNet) exhibit a greater distributional shift relative to the original training set (ImageNet-1K) than those in the internal test set.

## 5 DISCUSSION

From the experiment chapter, we can infer that a more sophisticated embedding model is more likely to have a better Confidence Curve (or higher *Normalized Confidence Gain*). With "sophisticated" we mean a larger number of dimensions, larger size of the entering image, and a richer dataset on which the model was trained. In particular, the dimensions of the images and the capacity of the embedding model significantly determine the performance.

We showed that combining embeddings from multiple models can be a promising strategy for integrating diverse representation spaces. However, the *Normalized Confidence Gain* obtained from

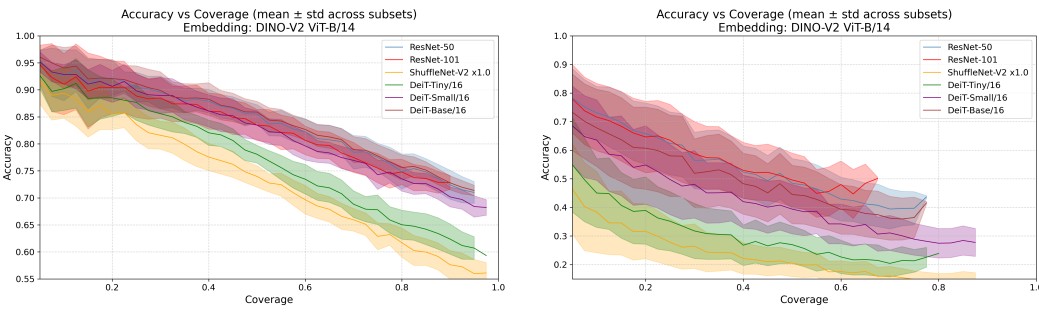

(a) Confidence Curves for Internal Dataset.   (b) Confidence Curves for External Dataset.

Figure 3: Accuracy vs Coverage (Confidence Curve) for the best Embedding Model ( DINO-V2 ViT-B/14 ) .

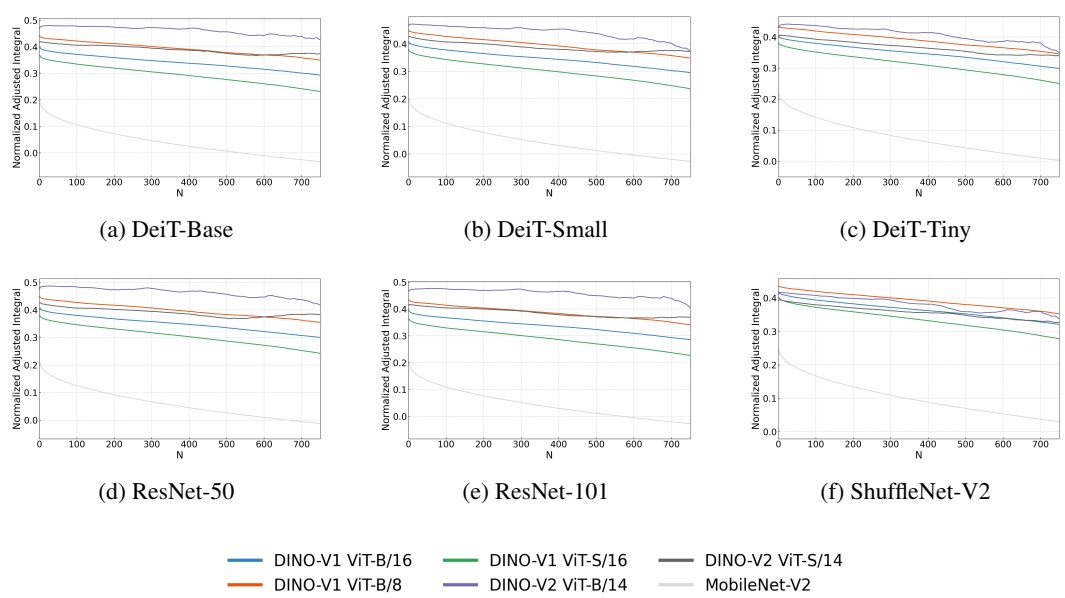

Figure 4: Normalized Confidence Gain vs number of neighbors $N$ for different classification models.

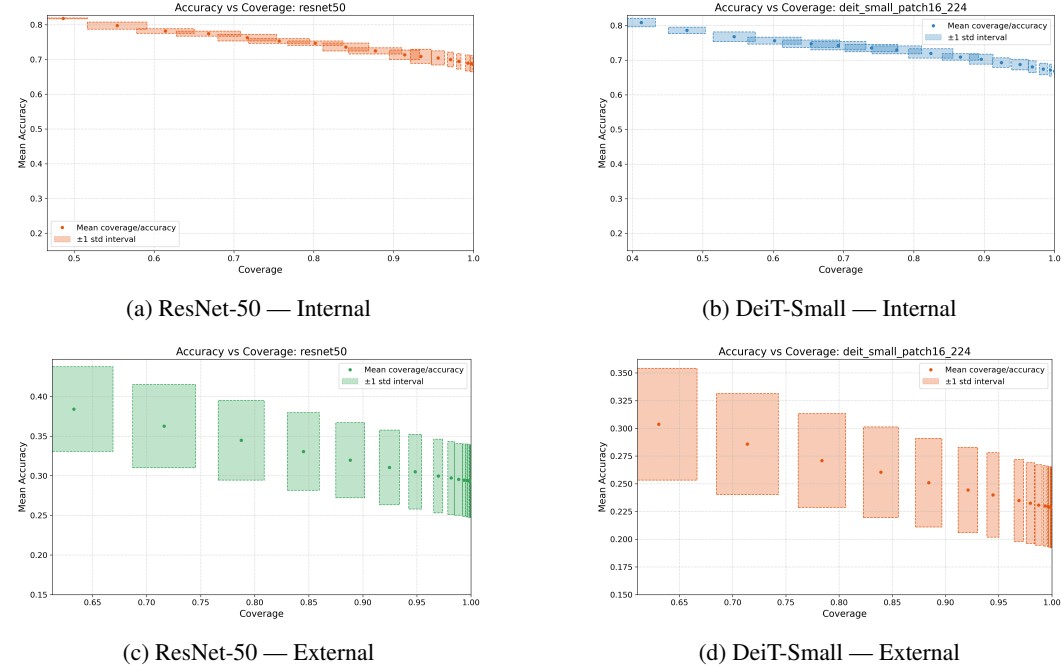

Figure 5: The Confidence Curve (Accuracy vs Total coverage) for the internal (**a**,**b**) and external (**c**,**d**) datasets for models ResNet-101 (**a**,**c**) and DeiT-Small (**b**,**d**). using the combinations of the embedding models.

such combinations was lower than that of the best performing individual embedding model. Unlike classical ensemble methods, where aggregating weaker models often leads to improved performance, we did not observe a performance gain in this setting. We identify two possible reasons for this. First, our combination strategy was a simple greedy approach and more sophisticated methods could yield better results. For example, a meta-classifier could be trained to select the most suitable embedding model for confidence estimation on a per-sample basis. Second, most of the models in our study were trained on only two distinct datasets. We expect that training on more diverse datasets

could enrich the combined confidence estimates. However, achieving such diversity is challenging, as off-the-shelf models are typically trained on large, overlapping datasets, limiting the potential for complementary knowledge.

We can see that after training, most of the models achieved the highest *Normalized Confidence Gain* with $N = 1$. Only the best-performing embedding models achieved the maximum value at significantly larger values of $N$ ($N > 1$). We believe that the more complicated the model, the larger the optimal value of $N$.

Our experiments were aimed solely at the image classification model. However, we suggest that a similar approach can work for other areas of Machine Learning, particularly for Natural Language Processing. But there is a caveat: we do not know how to partition the data efficiently. Images are distinct entities, and it is logical to compute the embeddings for each image. How exactly to divide the text corpora is unclear. There are two main questions in this investigation. What size of window to use and what jump step to use? Perhaps we need to have embedding collections of different window sizes, which could enrich the understanding of the neighborhood locality. Regarding the size of the jump, it would be interesting to calculate the embeddings with a step of one token. However, it would be computationally very heavy. In fact, English Wikipedia contains approximately 13.5 billion token instances across all articles, according to the TokTrack dataset (Flöck et al., 2017) (as of 2017). The training corpora used for large language models (LLMs) are estimated to be two to three orders of magnitude larger, although the exact size is unknown. For example, whereas GPT-3 was trained on approximately 499 billion tokens (Brown et al., 2020). In comparison, ImageNet-1K (used in this paper) contains approximately 1 million images and, correspondingly, 1 million embedding vectors. Moreover, it remains unclear how much efficiency is lost in neighborhood locality calculations when the step size exceeds a single token.

The current framework does not directly improve the performance of the classification model (or potentially other models). However, it enables the identification of samples that are out-of-distribution with respect to the original training set, thus making the model more confident in certain predictions.

# 6 CONCLUSION

In this paper, we showed that knowledge of the training dataset and its embeddings can be beneficial for determining a model's confidence in its predictions on newly seen data. The presented framework provides an alternative way to estimate prediction confidence. In particular, we significantly improved the accuracy of classification models without additional retraining—only by adjusting the number of neighbors—while skipping a minimal number of images that lie far from the training distribution. We demonstrated that using a single embedding model to filter out low-confidence samples can improve overall accuracy. Finally, we propose that the framework is not limited to object recognition and computer vision tasks, but could also be applied to other sensitive areas, such as Natural Language Processing, where hallucinations are common. To the best of our knowledge, this work does not present any ethical issues.

## REPRODUCIBILITY STATEMENT

All data used in this study are publicly available: ImageNet-1K, ImageNet-V2, and ObjectNet. The embedding models used in our experiments are also publicly available and were not retrained. The classification models evaluated for confidence and accuracy are likewise publicly available and were not retrained.

We provide an anonymized `.zip` archive in the supplementary materials containing: (i) all source code for the embedding extraction, confidence-curve calculation, and evaluation, (ii) instructions and configuration files to reproduce every experiment, and (iii) scripts to regenerate all tables and figures in the paper.

Running the provided scripts on the stated datasets reproduces every reported metric and plot.

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

## A  CONFIDENCE GAIN DETAILS

In this section, we explain in detail how the *Confidence Gain* is computed, alongside its key properties. Figure 1(a) illustrates the confidence curve for the [ResNet-101 & DINO-V2 ViT-B/14] model–embedding pair; however, the method applies to all model–embedding combinations.

We can observe that the distribution of points along the curve is not uniform: for large coverage values, there are almost no points, whereas for small coverage values, the points are densely packed. This is because a small coverage corresponds to more confident predictions, which occur more frequently. To handle the lack of points near full coverage (that is, near $coverage = 1$), we linearly extrapolated the confidence curve using the 20 points with the highest value of $coverage$. Furthermore, we noticed that for small values (below values of 0.1), the confidence curve tends to behave chaotically. Therefore, to ensure robustness, we computed the *Confidence Gain* by integrating over the interval $coverage \in [0.1, 1]$. Empirically, we observed that for $coverage > 0.1$, the accuracy behaves monotonically with the coverage and is less sensitive to noise.

We define the *Confidence Gain* as the area between the confidence curve (including extrapolation when necessary), the vertical line coverage $= 0.1$, and the horizontal line accuracy $= acc_b$. The $acc_b$ is the accuracy of the classification model when using 100% of the data, i.e., without any coverage-based filtering. This value is intrinsic to the classification model and the dataset and does not depend on the embedding model. As coverage $\rightarrow 1$, the accuracy along the confidence curve naturally approaches $acc_b$. We calculate the area using the `Scikit-Learn` library (Pedregosa et al., 2011).

We define the *Maximum Confidence Gain* as the area bounded by the four lines:
$x = 0.1$,
$x = 1.0$,
$y = 1.0$,
$y = acc_b$.

The value of the *Confidence Gain* is always less than the *Maximum Confidence Gain*. In other words, the Maximum Confidence Gain is a theoretical upper bound for the *Confidence Gain*. Therefore, it is logical to introduce the *Normalized Confidence Gain* as a ratio between these two values. The *Normalized Confidence Gain* would be determined in the range $[0, 1]$, where the value 1 corresponds to the perfect Confidence Gain, and the value 0 corresponds to no gain at all.

## B    LLM USAGE

This manuscript's text was polished using the large language model ChatGPT (GPT-4, May 2025 release). The model was used only for grammar and style improvements. All scientific ideas, experiments, analyses, and conclusions were conceived and verified by the authors, who take full responsibility for the final content.

