# OpenReview forum: "Prior Distribution and Model Confidence"
_ICLR.cc/2026/Conference — ICLR 2026 Conference Withdrawn Submission_

### Official Review · Reviewer_MMB8 · 2025-10-30

**Soundness:** 1
**Presentation:** 2
**Contribution:** 1
**Rating:** 0
**Confidence:** 4

**Summary:**

### Problem:
- Hallucinations in AI are widespread problem
- Studying hallucinations from a data-centric perspective seems promising, especially by leveraging latent space representations of training data. This does not require modifying or retraining models
- The term ‘hallucination’ is typically used in the context of generative language models. However, constructing datasets big enough for NLP is challenging. P

## Solution:
- The authors propose to use computer vision models as a test-bed to understand the causes of hallucinations
- The paper introduces a model-agnostic framework that estimates confidence using the distance of new samples from the training data in a learned embedding space.
- Their framework has the following steps:
    - Embedding distance: The authors compute embeddings of the training dataset using self-supervised and supervised vision models (DINO-V2 and MobileNet-V2). For each test image, they measure its distance to its nearest neighbors in this embedding space.
    - Confidence filtering: Predictions are accepted only if the test image lies close to a certain number of examples from training data. A threshold is introduced.
    - Multiple embedding models: The authors also combine multiple embedding models to capture different aspects of the training distribution.

**Strengths:**

- Important problem in real-world AI
- Use of both supervised and self-supervised embedding models

**Weaknesses:**

- The paper discusses LLM hallucinations, but then studies computer vision. This is a big leap. It’s not clear that this idea could be as easily applied to LLMs.
    - One issue is because of information leakage. How do I know that my test set wasn’t actually used in the LLM’s training data? This whole idea breaks down
- Severe lack of novelty. Embedding-based distance has been studied many times over. See below
    - Weak experiments: no comparison to existing methods
    - The paper mentions several related works (ODIN, Mahalabis OOD, Information theoretic OOD). Why do they not compare?
    - Other baselines:
        - Jiang et al. 2018. To Trust Or Not To Trust A Classifier
        - Papernot et al. 2018. Deep k-Nearest Neighbors: Towards Confident, Interpretable and Robust Deep Learning
        - In NLP: Khandelwal et al. ICLR’2020
- Hyperparameters $L_threshold$ and $N$. No hyperparameter study. Difficult to choose these in practice.

**Questions:**

The authors really need to do more comparisons to alternative techniques, especially since this domain has been so widely studied.
They also need to do something to demonstrate that the computer vision to NLP gap can be bridged by this technique

---

### Official Review · Reviewer_XeZ9 · 2025-10-31

**Soundness:** 3
**Presentation:** 2
**Contribution:** 2
**Rating:** 2
**Confidence:** 4

**Summary:**

The paper leverages recent progress in high-quality embeddings, learned independently of the target training set (via self/unsupervised learning or transfer learning), in order to assess a trained model’s confidence without retraining. The main idea is to estimate data density in the embedding space: for a new image, if too few similar training images lie nearby, the prediction is flagged as low-confidence and skipped. This simple filter is shown to raise overall test accuracy by avoiding likely mistakes. The authors evaluate several vision models and embedding extractors, finding that stronger embeddings work best; combining multiple embeddings can help but doesn’t always beat the best single one. Overall, the approach is claimed to be model-agnostic and also useful for spotting likely out-of-distribution cases.

**Strengths:**

The paper starts from the hypothesis that OOD detection in independent embedding spaces correlates well with a trained classifier’s confidence; its focus is the empirical evaluation of this hypothesis.

By using external, high-quality embedding models (e.g., DINO-v2), the method provides a robust confidence measure that is independent of the classifier’s internal biases and overconfidence.

**Weaknesses:**

Main weakness: the underlying hypothesis is already well established in the OOD literature, so the novelty is limited.

There is a body of literature on classification with a reject option [5], which is essential when using confidence to gate predictions. Appropriate metrics exist for evaluating such procedures, but they are neither discussed nor employed in this paper.

Additional points:

1. Missing Complexity Analysis: The paper does not include any analysis of the method’s time or space complexity, leaving its computational scalability unclear.

2. No Comparison to Fine-Tuning: The framework is not quantitatively compared against a minimally fine-tuned model. This omits the essential trade-off between the method's computational overhead and the performance gain achieved by a common alternative.

3. No Comparison to Established OOD Detection or Confidence Calibration Methods. The method effectively operates as a density-based OOD detector, yet it is not compared to widely accepted OOD baselines such as: (i) ODIN [1] which enhances the Maximum Softmax Probability baseline via temperature scaling and small input perturbations. (ii) Mahalanobis Distance-based OOD Detection [2] which models class-conditional feature distributions and measures Mahalanobis distance for OOD detection. (iii) Energy-based OOD Detection [3] which uses the energy score of the network’s logit outputs to distinguish in- and out-of-distribution samples.

4. Ambiguous Parameter Definitions: The critical distance threshold L-threshold, a central component in determining whether a sample lies within or outside the prior distribution, is not explicitly defined or accompanied by a quantitative rule for selection. The absence of a formula, heuristic, or sensitivity analysis limits reproducibility and interpretability.

5. Unclear Algorithmic Presentation: The main algorithm is described in a verbose and fragmented manner across multiple sections, without a concise pseudocode or structured workflow. This presentation makes it difficult to follow or reimplement the procedure. Moreover, the workflow diagram (Figure 1) adds visual complexity rather than clarity, as it does not clearly map to a step-by-step algorithmic sequence.

References
1. Liang, S., Li, Y., & Srikant, R. (2018). Enhancing the Reliability of Out-of-Distribution Image Detection in Neural Networks (ODIN). ICLR 2018.
2. Lee, K., Lee, H., Lee, K., & Shin, J. (2018). A Simple Unified Framework for Detecting Out-of-Distribution Samples and Adversarial Attacks. NeurIPS 2018.
3. Liu, W., Wang, X., Owens, J., & Liang, Y. (2020). Energy-based Out-of-Distribution Detection. NeurIPS 2020.
4. Guo, C., Pleiss, G., Sun, Y., & Weinberger, K. Q. (2017). On Calibration of Modern Neural Networks. ICML 2017.
5. Geifman, Y., and El-Yaniv R. (2019). Selectivenet: A deep neural network with an integrated reject option. ICML 2019.

**Questions:**

The results omit standard confidence and calibration metrics, such as Expected Calibration Error (ECE) [4] Brier Score, or Negative Log-Likelihood.  Without such comparisons, it remains unclear whether the approach truly enhances confidence estimation or calibration quality.

---

### Official Review · Reviewer_hihq · 2025-10-31

**Soundness:** 2
**Presentation:** 2
**Contribution:** 2
**Rating:** 2
**Confidence:** 4

**Summary:**

The paper studies selective prediction for image classification without modifying the trained classifiers. It builds a base embedding database from ImageNet-1K training images using pretrained embedding models (e.g., DINO-V1/V2 and MobileNet-V2). For a test image, the system retrieves nearest neighbors from the database and accepts the prediction of the classifier only if at least N neighbors meet the similarity threshold L_threshold; otherwise, it abstains. Adjusting acceptance criteria produces an Accuracy-Coverage (Confidence) curve. The paper introduces Confidence Gain and Normalized Confidence Gain. N is chosen on an ImageNet-V2 ''adjusting'' split to maximise NCG and then fixed for internal (ImageNet-V2 test split) and external (ObjectNet) evaluations. Overall, the method improves effective accuracy without retraining by leveraging proximity to the training distribution to guide abstention.

**Strengths:**

1. Model-agnostic confidence: Uses only proximity to the training distribution in embedding space—no reliance on the classifier’s own scores.

2. Systematic evaluations: Experiments cover multiple classifiers (ResNet-50/101, DeiT-Tiny/Small/Base, ShuffleNet-V2) and multiple embeddings (DINO-V1, DINO-V2 ViT-S/14, ViT-B/14, MobileNet-V2).

3. Consistent trend: Across most backbones, DINO-V2 ViT-B/14 gives the highest NCG; simple greedy combos of embeddings don’t beat the best single embedding.

4. Simple tuning: Only N is selected via NCG on the adjusting split and then reused.

**Weaknesses:**

1. Lack of external baselines. No comparisons with standard selective/OOD methods (ODIN, Mahalanobis, energy). Results are interpreted mainly via the paper’s own NCG and curves.

2. Threshold under-specification. The decision uses L_threshold, but only N is tuned on the ImageNet-V2 adjusting split; there’s no procedure or sensitivity for L_threshold.

3. Neighbour pool & scalability untested. Retrieval uses a fixed candidate pool (e.g., 1000) with no ablation on pool size vs. coverage–accuracy, and quantification of latency, index growth, or incremental updates; ANN backends (FAISS/ScaNN/HNSW) aren’t evaluated. This limits deployability claims.

4. No accuracy-at-coverage reporting. Tables provide NCG and baseline full-coverage accuracy (acc_b), but not accuracy at fixed coverage levels (e.g., 30%, 50%), which would aid operational decisions.

5. Scattered definitions. Formal details for NCG (integration over 0.1, 1, linear extrapolation near 1) are split across sections; consolidating formulas would help reproducibility.

**Questions:**

1. Could you report results on ImageNet-V2 (adjusting/test) and ObjectNet against ODIN, Mahalanobis, and energy-based scoring, using common selective metrics (e.g., AURC/EAURC) in addition to your NCG? This would contextualise the gains beyond intra-framework comparisons.

2. You explicitly tune N on the adjusting split but do not detail a procedure or sensitivity study for L_threshold. Can you
(a) specify how L_threshold is set,
(b) provide a 2D sensitivity plot over (N, L_threshold),
(c) state whether the same L_threshold generalizes across backbones/datasets?

3. Retrieval uses a fixed candidate pool (e.g., top-1000) prior to the N/L_threshold test. Please ablate the pool size (100/500/1000/2000) to show its effect on the coverage–accuracy curve and NCG.

4. Beyond NCG and the baseline full-coverage accuracy (acc_b), can you provide accuracy at a few operational coverage levels (e.g., 30%, 50%, 70%) for the main models? This would make the trade-off immediately interpretable for deployment.

5. Results show DINO-V2 ViT-B/14 often achieves the highest NCG. Could you quantify how much of this comes from embedding capacity vs. pretraining data differences (e.g., V1 vs. V2)? A controlled comparison (matched dimension/resolution when possible) would clarify the driver of gains.

---

### Official Review · Reviewer_b6kt · 2025-11-01

**Soundness:** 2
**Presentation:** 3
**Contribution:** 2
**Rating:** 4
**Confidence:** 4

**Summary:**

This paper proposes a framework for estimating model prediction confidence based on the distribution of training data in the embedding space. By computing embeddings of training samples (using pre-trained models such as DINO or MobileNet), the authors filter out low-confidence predictions according to their distance to the training distribution. The method does not require model retraining and reportedly improves classification accuracy on ImageNet-V2 and ObjectNet.

**Strengths:**

1. The paper tackles an important question — how the training data distribution affects model confidence and reliability.

2. The experimental setup is transparent and uses public datasets and pretrained models.

3. Results across CNNs (ResNet, ShuffleNet) and Vision Transformers (DeiT) provide a broad view of the observed phenomenon.

**Weaknesses:**

1. The method is essentially a nearest-neighbor–based confidence filtering strategy in embedding space. Similar concepts have been extensively explored in prior work on OOD detection and confidence estimation (e.g., Mahalanobis distance, energy-based OOD detection, influence functions, etc.). The contribution is incremental and not theoretically or algorithmically new.

2. The proposed “Normalized Confidence Gain” metric is a simple normalized area under an accuracy–coverage curve, but the paper does not provide any theoretical justification or comparison to standard uncertainty metrics such as AUROC, ECE, or NLL.

3. (a) The improvements reported are small and inconsistent, especially on the external test set (ObjectNet).
    (b) There is no comparison to strong baselines such as ODIN, Energy-based OOD, OpenOOD, or ensemble calibration methods.
    (c) The “embedding combination” experiment fails to improve results, which undermines one of the paper’s main claims.

4. The paper claims to address hallucination detection, but the actual experiments are standard image classification filtering tasks. The connection to hallucination phenomena in generative models is speculative and unsupported.

5. The paper does not explain why some embedding models yield better confidence estimation or analyze the effect of key parameters such as neighborhood size N. No computational cost or scalability discussion is included.

6. The paper is overly long relative to its conceptual simplicity, with repetitive discussion and speculative remarks (e.g., about NLP extension) without experimental grounding.

**Questions:**

1. How does this method compare to established OOD detection or uncertainty estimation methods (e.g., ODIN, Energy-based, Mahalanobis, Deep Ensembles)?

2. What is the conceptual or empirical advantage of the “Normalized Confidence Gain” metric over standard metrics such as AUROC or calibration error?

3. Why do results on ObjectNet show minimal improvement? Does this indicate the approach is only effective for in-distribution filtering?

4. What is the computational complexity of nearest-neighbor search at ImageNet scale? Is the method practical for large models or datasets?

5. If the approach is intended to generalize to NLP, can the authors provide a concrete prototype or preliminary experiment?

---

### Note · Authors · 2025-12-09

**Comment:**

We respectfully request withdrawal of this submission due to planned revisions. Thank you for your consideration.

**Withdrawal Confirmation:**

I have read and agree with the venue's withdrawal policy on behalf of myself and my co-authors.